# Pregnant Mothers’ Medical Claims and Associated Risk of Their Children being Diagnosed with Autism Spectrum Disorder

**DOI:** 10.3390/jpm11100950

**Published:** 2021-09-24

**Authors:** Genevieve Grivas, Richard Frye, Juergen Hahn

**Affiliations:** 1Department of Biomedical Engineering, Rensselaer Polytechnic Institute, Troy, New York, NY 12180, USA; grivag@rpi.edu; 2Center for Biotechnology and Interdisciplinary Studies, Rensselaer Polytechnic Institute, Troy, New York, NY 12180, USA; 3OptumLabs Visiting Fellow, OptumLabs, Eden Prairie, MN 55344, USA; 4Department of Child Health, University of Arizona College of Medicine, Phoenix, AZ 85004, USA; rfrye@phoenixchildrens.com; 5Phoenix Children’s Hospital, Phoenix, AZ 85016, USA; 6Department of Chemical and Biological Engineering, Rensselaer Polytechnic Institute, Troy, New York, NY 12180, USA

**Keywords:** autism spectrum disorder, medical claims, logistic regression analysis, retrospective analysis, associated risk

## Abstract

A retrospective analysis of administrative claims containing a diverse mixture of ages, ethnicities, and geographical regions across the United States was conducted in order to identify medical events that occur during pregnancy and are associated with autism spectrum disorder (ASD). The dataset used in this study is comprised of 123,824 pregnancies of which 1265 resulted in the child being diagnosed with ASD during the first five years of life. Logistic regression analysis revealed significant relationships between several maternal medical claims, made during her pregnancy and segmented by trimester, and the child’s diagnosis of ASD. Having a biological sibling with ASD, maternal use of antidepressant medication and psychiatry services as well as non-pregnancy related claims such hospital visits, surgical procedures, and radiology exposure were related to an increased risk of ASD regardless of trimester. Urinary tract infections during the first trimester and preterm delivery during the second trimester were also related to an increased risk of ASD. Preventative and obstetrical care were associated with a decreased risk for ASD. A better understanding of the medical factors that increase the risk of having a child with ASD can lead to strategies to decrease risk or identify those children who require increased surveillance for the development of ASD to promote early diagnosis and intervention.

## 1. Introduction

Autism spectrum disorder (ASD) is an early onset neurodevelopmental disorder characterized by difficulties in social communication/interactions and by the presence of restricted and repetitive behaviors [1]. The prevalence of ASD has significantly increased over the last three decades [2] with the most recent estimate being 1 in every 54 eight-year-old children in the United States has been diagnosed with ASD with a 4.3 times higher occurrence in males than females [3]. The etiological understanding of ASD has also changed over the years, with current research suggesting a combination of genetic and environmental factors [4]. It is now generally acknowledged that investigation of environmental risk factors for ASD should not only be limited to the life of the child, but also include the prenatal and preconception period [5].

Numerous maternal body systems have been hypothesized to contribute to ASD including the gastrointestinal, immune, metabolic, and endocrine systems [6,7,8,9,10]. It is not surprising that research has extended these investigations to include the influence of maternal systems disorders. Maternal endocrine or hormonal disorders, such as polycystic ovary syndrome, show an increased risk of offspring developing ASD [11]. Maternal autoimmune disorders are notable ASD risk factors [12,13,14,15] with emphasis on hypothyroidism [16], psoriasis [17], and rheumatoid arthritis [18]. The presence of maternal infection during pregnancy significantly increases the risk of ASD in the offspring [19], with studies suggest this effect is specific for bacterial [20], viral [21], severe [22,23] or febrile [24,25] infections. In fact, the maternal immune activation (MIA) mouse model, a major animal model of ASD, induced ASD-like behavior in the offspring by activating the material immune system but also highlights the variability of this effect [26]. Though it may provide difficult to distinguish the confounding effects of the infection itself from the treatment for the infection, as some studies have shown antibiotic consumption is a risk factor, though there are discrepancies regarding the significance for antibiotics taken during the second or third trimester [21] or when taken for longer than 14 days [27].

It is well known that ASD is commonly linked to other cognitive or mental health disorders such as epilepsy, ADHD, and anxiety [28,29,30,31], and the role of brain development cannot be understated. Similarly, the influence of maternal mental disorders is crucial to understand and has been widely studied, with a heavy focus on maternal depression and antidepressant usage during pregnancy. Recent literature suggests the risk factor for ASD may be associated with prior antidepressant treatment or maternal psychological conditions rather than antidepressant consumption [5,32]. Another widely studied maternal pharmaceutical is prenatal vitamin supplementation, particularly folate (vitamin B9), which has been found to reduce the risk of offspring developing ASD by almost half [33,34,35].

Lastly, risk factors have also been found for delivery-related events such as preterm delivery [36,37] and cesarean delivery [38,39]. Though, these factors may be influenced by abnormal child development stemming from previously mentioned risk factors.

Given the large number of studies that have presented contradicting results, this work focuses on identifying ASD risk factors from a very large cohort of mothers in the United States. Specifically, this study is a retrospective analysis of maternal medical events that occurred during pregnancy and their effect on the risk of ASD in the child. These maternal events are reflected by diagnostic, procedural, and pharmaceutical claims from a private United States health plan.

## 2. Materials and Methods

### 2.1. Mother and Child Cohort Identification

This retrospective analysis used de-identified claims data with a family identifier and socioeconomic status information from the OptumLabs^®^ Data Warehouse (OLDW), which included medical and pharmacy claims, laboratory results, and enrollment records for commercial and Medicare Advantage (MA) enrollees. The database contained longitudinal health information on enrollees and patients, representing a diverse mixture of ages, ethnicities and geographical regions across the United States [40]. As this study uses deidentified data, approval is exempt from the Institutional Review Board.

Children diagnosed with and without ASD, born between 1 January 2000 and 31 December 2010, were previously identified using the OLDW [10]. Vargason et al. (2019) used the children’s diagnostic claims from their date of birth until five years of age to identify children diagnosed with ASD. This study identified the mother of these children through the use of family identifiers, policy holder relationship codes, and delivery claims within 10 days of the child’s earliest enrollment date (assumed to be the child’s date of birth) [41,42]. Diagnostic, pharmacy, and procedural claims (Appendix A) were identified for each mother ten months prior to the birth of each child.

The processing steps used to identify children and their mothers, as well as the resulting number of women and children identified from the OLDW, are outlined in Table 1. Women were identified between the ages of 14 and 49 with commercial health coverage that included medical, pharmaceutical, and mental health coverage; this medical plan matched that of the children cohort. Step 4 outlined in Table 1 required that all children be labeled as “child” in relation to the policy holder. In Step 6, women were linked to children by having a delivery claim within 10 days of the child’s first enrollment date. During this process, some children were found to be linked to multiple mothers, most likely due to different women under the same policy having birth claims at similar times; these mothers and children were excluded in Step 7. The final cohort sizes, shown in Step 8, identified various pairs of siblings (different children having the same mother) during the investigated time frame, resulting in a greater total number of children than women. This included siblings (single and multiple births) with different ASD outcomes, i.e., with and without an ASD diagnosis. For this study, each child was associated with the events that occurred during his or her gestational period, which were unique for siblings but identical for multiple births (i.e., twins and triplets). The final data set was comprised of 123,824 pregnancies identified using the OLDW data base; 1265 pregnancies resulted in children with ASD (ASD cohort), and 122,559 pregnancies resulted in children with no ASD diagnosis (population or POP cohort) during the investigated time frame. The ASD prevalence determined in this study agrees with the prevalence estimation performed by the Center for Disease Control during the same time period [43]. Within the dataset, 37,775 (30.5%) of all children had a sibling and of this 5616 (4.5%) were a part of multiple births.

### 2.2. Medical Claims Identification

Medical claims were split up into the following three categories of variables: diagnostic claims identified by the International Classification of Diseases coding, version 9 (ICD-9), filled prescription claims determined by the National Drug Code (NDC) identifiers, and medical procedures claims denoted by their Current Procedural Terminology (CPT) coding. Total claims investigated included 478 ICD-9 diagnostic codes as variables, 10,810 NDC codes segmented into 132 pharmacy variables, and 3,808 CPT codes segmented into 122 procedural variables (Appendix A). Pharmacy variables were created based on code descriptions embedded into the database. Procedural variables were based on CPT code descriptions. Table 2 shows the progression of variable selection from these categories, further outlined below.

All variables had relatively similar percentages of claims between both cohorts (Appendix A). A heuristic threshold was used to exclude variables due to uncommon claims, thereby eliminating small cell sizes and ensuring that all variables were present for both cohorts. If the number of claims fell below 2% for both cohorts combined (2476 of 123,824) or 2% for the ASD cohort only (25 of 1265) then the claim was excluded from further analysis. An example can be found in Appendix A, depicting the number of pregnancies that had a claim for the first 100 diagnostic variables (Appendix A). Only variables with claims greater than 2% for the combined and ASD cohorts were further investigated (noted by the dashed lines in Appendix A, top and bottom, respectively). Thus, of the first 100 diagnostic variables, only 3 diagnostic variables were kept. Due to OptumLabs Data Policy, all cell values less than 11 are censored for de-identification purposes as noted by the *y*-axis starting at 11 for all figures.

Of the original 732 variables investigated, 156 remained after thresholding: 82 ICD-9 diagnostic codes, 27 pharmacy variables, and 45 procedural variables, see Table 2. These claims were then used to identify the mothers’ gestational ages, as well as associated trimesters, using the same protocol presented in Li et al. [44]. Claims that fell outside of the identified gestational days were then removed and the remaining variables with claims above the threshold were kept. In addition, the following 6 maternal sociodemographic variables were included: race, home ownership, education level, income level, age, and a binary indicator for women who have had previous children with ASD. The latter variable, denoted as ‘Previous ASD’, refers to all subsequent children whose mother had a previous child diagnosed with ASD during the time frame of this study. This variable was included because women who have had previous children with ASD are at an increased risk of having another child with ASD [45].

For comparison, the sociodemographic data were analyzed based on individual women (referred to as “women cohorts”) instead of pregnancies or resulting children (referred to as “pregnancy cohorts”) to better represent the population. For this case, cohort separation is defined as women who have never had a child with ASD (population or POP) and women who have had one or more children with ASD (ASD). For ASD cohort women, age range refers to the age at which each woman had her first child with ASD, for POP cohort women it refers to the age at which she had her first child.

### 2.3. Statistical Analysis

A chi-square analysis was used to determine a statistically significant difference in proportions between the women’s ASD and POP cohorts for the sociodemographic variables. For small cell sizes, a Fisher’s exact test was used. For each age category, a Welch’s t-test was used to determine statistically significant differences between mean age of the ASD and POP cohorts.

An F-test, with 5% significance level, was used to determine a statistically significant difference in variance between the ASD and POP cohort for total number of medical claims (diagnostic, pharmacy, and procedural combined), as well as each variable category individually. For categories that showed a statistically significant difference in variance, a Welch two-sample *t*-test was used to determine a statistically significant difference between the mean number of claims for the ASD and POP cohorts, at a 5% significance level. For equal variance, a standard t-test was used at a 5% significance level. Histograms were normalized in order to better compare the two cohorts.

Logistic regression was used to estimate the relationship between the presence of ASD in the child over the investigated time frame and the maternal medical claim (diagnostic, pharmacy, or procedural) made during pregnancy [46]. Pearson correlation analysis was conducted to identify claims that were highly correlated with any other claim, *r* > = 0.7, of which those of lower significance (denoted using *p*-values calculated from the unadjusted logistic regression) were removed from the analysis (see Table 2) [47,48]. Adjusted odds ratios (ORs) were used to quantify the effect of the medical claim and the associated risk of the child being diagnosed with ASD later, using a 95% confidence interval [49]. An initial logistic regression model showed the previous ASD variable was highly skewed towards the ASD cohort due to bias associated with multiple births; to correct for this, two adjusted logistic regression models were used, one with all variables and one with all variables except previous ASD. The statistically significant variables determined from these two models were then used for a third adjusted logistic regression model, which allowed for correction of multiple comparisons by reducing the number of variables included in the model as well as identifying false significance from the latter models. A schematic of the model development can be found in Figure 1. Due to computational restrictions, all logistic regression models were built using 10% of the POP cohort data, resulting in a ratio of approximately 10:1 POP to ASD, stratified based on sociodemographic variables. This analysis was then repeated for each trimester excluding sociodemographic variables since these are constant throughout the entire pregnancy. For brevity, all statistically significant findings as defined in this section and reported in this study are referred to as significant.

Some diagnostic variables were further specified based on their ICD-9 coding. ICD-9 codes are structured through a numeric system where a whole value code can also contain decimal values to elaborate on a diagnosis. For example, ICD-9 code 649 corresponds to the variable Other Conditions Complicating Pregnancy and can be broken down as follows: 649.0 tobacco use disorder complicating pregnancy, 649.1 obesity complicating pregnancy, 649.2 bariatric surgery complicating pregnancy, etc. Claims are made using either whole values or decimal points, at the discretion of the medical professional. All statistically significant final diagnostic variables are further evaluated by their whole and single digit ICD-9 coding in a separate logistic regression analysis, shown in Figure 1. Variables were only included in this analysis if they contain claims greater than 11, and due to this smaller threshold were evaluated with both unadjusted and adjusted ORs.

## 3. Results

There were some key differences in sociodemographic data for the two women cohorts (Table 3). The percentages for race were for the most part comparable, with the majority being White, however, a significantly larger percentage of Asian pregnancies in the ASD cohort (10.4%) was found compared to the POP cohort (7.7%). In addition, the ASD cohort, as compared to the POP cohort, was significantly more educated (i.e., attained a degree higher than a Bachelor’s degree), had higher income (i.e., income greater than USD 125,000) and was older (i.e., age 30 years or older), see Table 3 and Table 4. Furthermore, the ASD cohort had a significantly smaller percentage of women having only a high school diploma, an income between USD 40,000–74,999, and being of age between 20–29 years old. The ASD cohort had a significantly higher percentage of previous ASD children, 245 (20.1%), compared to the 0 (0.0%) from the POP cohort. This was obviously expected since the women POP cohort is defined as women who have never had a child with ASD and thus will not have a previous ASD indicator.

Correlation analysis for all variables during the entire pregnancy depicted six pairs of variables containing correlations of 0.7 or higher. These variables and their associated unadjusted *p*-values are shown in Appendix A. The variable with the larger *p*-value in each pair-wise correlation was discarded from the adjusted logistic regression analysis; variables that remain in the analysis are bolded in Appendix A. Three pairs of highly correlated variables all related to the same medical event of receiving a vaccination (variables: Vaccinations, Need for Prophylactic Vaccination against Viral Diseases, and Immunization Administration for Vaccinations). The remaining three pairs of variables were associated with a cardiovascular procedure, diabetic-related materials (such as test strips), and a thyroid disorder.

Normalized histograms for claims from both the ASD and POP cohort can be found in Figure 2 with associated descriptive statistics in Table 5. These data were generated by summing claims made throughout each entire pregnancy. Histograms for all medical claims (diagnostic, pharmacy, and procedural) and only diagnostic claims (Figure 2A,B, respectively) closely followed a normal distribution as shown by the similar mean, median, and mode values listed in Table 5. Pharmacy claims and procedural claims (Figure 2C,D) were right- and left-skewed, respectively, where most women had few (1–2) prescriptions and many (14–16) procedural claims. For all categories of variables (including total combination), the ASD cohort had a statistically significantly higher mean number of claims than the POP cohort.

Normalized histograms for claims made in each trimester, along with descriptive statistics, can be found in Figure 3 and Figure 4, as well as Table 6. The largest number of total medical claims was made in the third trimester with similar values for the first and second trimester. While the mean and median number of claims was greater in the first trimester compared to the second, the first trimester had a greater amount of zero-claims. In all trimesters, the majority of claims were made for procedures, followed by diagnostics and pharmacy. For all trimesters, the ASD cohort had a significantly higher mean number of diagnostics, pharmacy and procedural claims with one exception: diagnostic claims made in the first trimester showed no significant difference.

The adjusted logistic regression models (with and without previous ASD) show a total of 20 significant variables (Appendix A): 2 sociodemographic variables, 7 diagnostic variables, 3 pharmacy variables, and 8 procedural variables. When modeled by themselves, only 17 of these significant variables retained their significance (Table 7, full model results can be found in Appendix A). A majority of the variables (13 of 17) were associated with an increased risk of having a child diagnosed with ASD. While both sociodemographic variables showed an increased risk, having a child previously diagnosed with ASD was associated with the largest increased risk of all variables, OR 16.09 (8.27, 32.12). Three diagnostic, all three pharmacy, and five procedural variables were also associated with an increased risk. The remaining four variables (two diagnostic and two procedural) were associated with a significantly decreased risk. Results for the subcode logistic regression analysis on diagnostic variables from the entire pregnancy can be found in Appendix A.

Results from the adjusted logistic regression analyses for all trimester variables can be found in Appendix A. The final significant variables identified for each trimester can be found in Table 8; some of these variables differed from the entire pregnancy analysis due to the different number of claims that occur in each trimester. A larger number of significant variables occurred for the first and third trimesters (15) compared to the second trimester (13). While all trimesters had a majority of variables associated with increased risk, the third trimester had the most variables associated with increased risk (11) while the first trimester had the most variables that were associated with decreased risk (5). Multiple Gestation, Antidepressants, and Procedure Services Psychiatry variables were consistently associated with increased risk for all three trimesters.

The variable Other Conditions Complicating Pregnancy was associated with a significantly increased risk during the third trimester, however further analysis showed that no subcode was significant for this occurrence and thus this study is not able to determine what event influenced this diagnosis (see Appendix A). Similarly, Services Office or Other Outpatient was associated with a significantly increased risk during the second trimester. This variable corresponds to a new patient visit; however, this study is unable to determine if this visit was related to pregnancy or another maternal health-related event.

## 4. Discussion

The majority of this study’s significant findings were associated with an increased risk of having a child with ASD. Many of these correspond to a single variable in the model such as having a previous child with ASD (Table 7, Previous ASD), first pregnancy over the age of 35 (Table 7, Other Indications Related to Labor), current cesarean delivery (Table 7 and Table 8, Other Complications of Labor), prescription for antidepressants (Table 7 and Table 8), psychiatric services (Table 7 and Table 8), pre-existing diabetes (Table 7, Durable Medical Equipment Diabetic), urinary tract infection during the first trimester (Table 8, Other Disorders of Urethra and Urinary Tract), and premature pregnancy (Table 8, variables Normal Pregnancy and Surgical Procedures Maternity Care and Delivery). Some of these variables were grouped in order to identify a common theme associated with ASD such as variables corresponding to standard obstetrical procedures, non-pregnancy related procedures, or maternal immune dysfunction and allergens. Lastly, a few variables and their associations with ASD disagreed with current literature, such as being of Asian race, having a prescription for pre-natal vitamins, and having multiple gestations. These findings, and others, have all been further discussed below and a summary can be found in Table 9, listed as they appear in this section.

The data cohorts identified in this study found the highest percentage of ASD pregnancies among White children, followed by Asian, Hispanic, and Black children. This trend agrees with that reported in a CDC surveillance completed within the same time period as this study [43] except for those of Asian race, which was found to vary widely depending on location and where our study shows a significantly larger proportion in the ASD women’s cohort (*p*-value < 0.001, Table 3). Asian race was associated with a 40% increased risk of having a child diagnosed with ASD (Table 7). The most recent CDC surveillance summary showed similar prevalence between Asian and White children within the United States [3]. Therefore, it is most likely that the increase in risk associated with Asian race found in this study was a result of sample bias as noted by the significantly greater percentage of Asian ASD women identified in Table 3.

It is well known that the recurrence risk for ASD in families is much greater than the risk for the general population, therefore, women who have a child with ASD are considered high-risk for having subsequent children diagnosed with ASD [45]. Our study confirmed and clarified this finding, suggesting a 16-fold increased risk associated with having another child with ASD when a previous child was diagnosed with ASD (Table 7). There was also evidence to suggest that this elevated risk increased with each additional child diagnosed with ASD [50,51]. However, having a child diagnosed with ASD may influence the parental decision of having subsequent children, known as reproductive stoppage, which is a confounding factor [51,52,53].

Sociodemographic trends for ASD noted a higher prevalence of the disorder among higher levels of education and income [54,55], similar to what was found in this study (see Table 3). While advanced maternal age was not reflected in the sociodemographic variables of the logistic regression analysis, it was reflected in the diagnostic subcode Elderly Primigravida (ICD-9 659.5, Appendix A) of the variable Other Indications Related to Labor, ICD-9 659, which demonstrated an overall adjusted OR 1.28 (1.13, 1.45; Table 7). This subcode corresponds to women with their first pregnancy over the age of 35. Advanced maternal age has been associated with an increased risk of ASD, with studies showing the association for the highest age category [56], age greater than 35 [5,57], or age greater than 40 [58].

Our analysis showed that cesarean delivery (ICD-9 669.7, Appendix A) was the significant contributing factor to the diagnostic variable Other Complications of Labor, ICD-9 669, OR 1.27 (1.09, 1.47), Table 7, overall as well as during the third trimester, OR 1.22 (1.05, 1.42), Appendix A. Previous studies reported inconclusive results for associating ASD with cesarean delivery. Some studies showed a weak or no association [15,59,60] while others showed a significantly increased risk [39,61] though this may be correlated with the risk factors associated with the cause for cesarean delivery instead of the delivery itself [62,63]. Some studies find cesarean delivery with general anesthesia significantly increased the risk of ASD compared to cesarean delivery with regional anesthesia or other indications [38,64]. While some women elect to have a cesarean delivery, more commonly they occur due to complications that arise during pregnancy or delivery, which vary depending on maternal age [65]. Cesarean deliveries change the risk profiles for both the mother and newborn [66] and may directly affect the environment of newborns and possibly even their microbiome [67,68]. There are even long-term health risks associated with the delivery following a cesarean [69]. However, our study found that having a previous cesarean delivery (before the current pregnancy), code ICD-9 654.2, was associated with a decreased risk of ASD during the third trimester (Appendix A), with an overall OR 0.86 (0.75, 0.99) denoted by Abnormality of Pelvis, ICD-9 654, Table 8. While currently there is little research on the effect of previous cesarean delivery or even vaginal birth after cesarean (VBAC) and having a child diagnosed with ASD, women with previous cesarean deliveries are more carefully managed, especially during labor [70,71]. It is possible that these extra precautions are a confounding factor as this finding contradicts other literature that associate prior cesarean delivery with an increased risk of adverse reproductive outcomes for subsequent pregnancies [72,73,74,75].

A prescription for antidepressants was significantly associated with an increased risk for ASD, with an overall risk greater than 40% and increased per trimester. Maternal antidepressant usage is a highly researched area as a potential risk factor for ASD. Many studies have shown that antidepressants, including the use of selective serotonin reuptake inhibitors (SSRIs), are associated with a significantly increased risk of the child developing ASD [5,76,77]. Contrary to this, other reviews find conflicting results [78,79] or no significant association [80]. However, recent investigations also examined the underlying mental illness, as many studies have shown that adjusting for depression attenuates the significant association of antidepressants while the association of mental illness remains strong [81,82,83,84,85,86]. Our analysis found a greater than 40% significantly increased risk associated with psychiatric services (Table 7), that increased to greater than 60% during the first trimester and decreased with each trimester (Table 8). We did not find a sufficient number of claims for Major Depression Disorder (ICD-9 311) to include in this analysis. As antidepressant medications are widely used for multiple psychiatric conditions, including anxiety, bipolar disorder and others, the data may suggest that an increased risk of ASD may be associated with a wider array of psychiatric conditions in the family, as have been documented in other studies [87]. However, it is clear that the risk associated with maternal antidepressant usage is heavily influenced based on study design [88], and that it is of great importance to acknowledge the underlying confounding effects of mental health disorders.

Our study did not have a sufficient number of claims to include the diagnosis of diabetes (ICD-9 250) and did not find a diagnosis for gestational diabetes (ICD-9 648) significant, but found that a prescription for Diabetic Durable Medical Equipment (DME, such as insulin needles) was associated with a significantly increased risk of ASD, OR 1.27 (1.00, 1.59), Table 7. This finding suggests a significant association with diabetes that may have been diagnosed before the time of our study. The risk of maternal diabetes associated with ASD remains unclear, with reviews suggesting a strong [89,90], moderate [15], or no [91] relationship. While some studies combine the effects of any type of diabetes, others suggest that familiar type 1 diabetes [13], gestational diabetes [92,93], or only diabetes in conjunction with obesity [94] are associated with an increased risk. Though, individuals who are predisposed to diabetes may act as a confounding factor. Extensive reviews have been conducted on how diabetes may relate to biological mechanisms involved in the development of ASD, specifically through the oxidative stress pathways [95].

Many studies show a significantly decreased risk associated with pre-natal vitamins [33,96], such as folic-acid supplements [34,97] or fatty acids [98]. However, our study showed an increased risk, overall OR 1.18 (1.04, 1.33), shown in Table 7, as well as a similar increased risk in the second and third trimesters, Table 8. This finding may not truly reflect the relationship between pre-natal vitamins and ASD as most vitamins are provided over the counter and therefore do not appear within an insurance claim and are not represented in this study. It is also possible that there are unknown reasons associated with receiving a prescription for nutritional vitamins that may be acting as a confounding factor such as economic concerns or medical conditions. For example, individuals with a previous child with ASD may specifically request a prescription for vitamins. The fact that a large well-done study demonstrated that higher folate supplementation was associated with a decreased risk of ASD would suggest other confounding factors are possible [35]. In addition, our study did not quantify the type of vitamins prescribed, (i.e., vitamin D, vitamin B, multivitamin, dietary supplements, etc.). For example, prescribing folic acid, an oxidized folate that is poorly metabolized and poorly transported across the placenta in some women, as opposed to a reduced folate which has much higher bioavailability, can result in high levels of unmetabolized folate in the blood in those with poor folate metabolism [99]. This can lead some to make the wrong conclusions that too much folate supplementation during pregnancy can be associated with an increased risk for ASD [100], whereas the problem lies with providing the correct type of bioavailable folate [101].

One finding that did show a significantly decreased risk of having a child with ASD, was a procedure claim for receiving a vaccination, OR 0.58 (0.46, 0.71), Table 7. This was also significant in the third trimester, OR 0.68 (0.47, 0.96), as well as the second trimester (diagnosis claim Need for Prophylactic Vaccination against Certain Viral Diseases), OR 0.60 (0.41, 0.83), Table 8. This study was unable to determine the type of vaccination, however. There is a limitation of relating this paper’s findings to influenza vaccination as many instances of this vaccination can occur within the community, outside of a doctor’s office, and thus would not appear within the claim’s data. Vaccinations have been recommended during pregnancy in order to prevent infections [102].

Other standard obstetrical procedures showed a decreased risk, specifically Uterine Size and Date Discrepancy (Appendix A, ICD-9 649.6), Special Screening of Malignant Neoplasms (of Cervix) overall and during the first trimester (Table 7 and Table 8, respectively), Cervical Incompetence (ICD-9 654.5 Appendix A) during the first trimester, Antenatal Screening and Pregnancy Evaluation (ICD-9 V28.5-6 and V72.4, respectively, see Appendix A) during the first trimester, Diagnostic Ultrasound Procedures during the second trimester (Table 8), physical therapy (Evaluations Physical Medicine and Rehabilitation) overall and during the second trimester (Table 7 and Table 8, respectively), and Procedures Other Pathology and Laboratory during the first trimester (Table 8). It is well documented that obstetric complications increase the risk of having a child with ASD [103,104,105] and it is clinically recommended that women who are at high risk should be closely monitored throughout their pregnancy [102]. Thus, these findings suggest that women with earlier and more aggressive obstetrical care have a decreased risk of ASD.

Various hospital procedures showed an increased risk such as an in-hospital consultation (Services Consultation) overall and during the first and third trimester (Table 7 and Table 9, respectively), surgical procedures that may require anesthesia (Anesthesia Procedures Lower Abdomen shown in Table 7, Surgical Procedures Female Genital System during the first trimester and Surgical Procedures Nervous System during the third trimester shown in Table 8), ventilation or breathing tests (Procedures Pulmonary, Table 7), and Procedures Diagnostic Radiology (Table 7). While the following claims may not have required hospitalization they also show an increased risk: Special Screening for Blood Disorders (Table 7), Special Screening for Endocrine Nutritional Metabolic and Immunity Disorders during the first trimester, Abdominal Pain (Other Symptoms Involving Abdomen and Pelvis) during the first trimester, Acquired Hypothyroidism during the third trimester, Consultations Clinical Pathology during the first trimester, and Services Office or Other Outpatient during the second trimester (all of which can be found in Table 8). These findings suggest that claims not relating to the pregnancy nor delivery are associated with an increased risk of ASD regardless of trimester.

Our study showed an increased risk of UTI (Other Disorder of Urethra and Urinary Tract) but only during the first trimester, OR 1.49 (1.19, 1.86), shown in Table 8. UTIs have been shown to be common during pregnancy but have inconclusive associations with ASD [19,20,21,22,24,25,60,106]. Urinalysis procedures showed an increased ASD risk during the third trimester, OR 1.18 (1.04, 1.33), Table 8. However, urinalysis procedures refer to any urine examination and is not only associated with diagnosing UTI but may include other tests such as testing for pre-eclampsia. Pre-eclampsia has been shown to have an increased risk for ASD [5,107], but a diagnosis (ICD-9 642) was not found to be significant in this study. Maternal Antepartum Hemorrhage and Placenta Previa showed a decreased risk of ASD during the third trimester, OR 0.71 (0.55, 0.92), Table 8. Antepartum hemorrhage has been shown to be associated with intellectual disability but not ASD [60], while placenta previa is associated with a decreased incidence of pre-eclampsia [108].

Premature (pre-term) children, identified by the diagnostic variable Normal Pregnancy (ICD-9 V22) and the procedural variable Surgical Procedures Maternity Care and Delivery during the second trimester (Table 8), were associated with a 30% increased risk, consistent with Talmi et al. as well as other previous studies that found preterm to be a significant factor associated with ASD [37,109,110]. There is a higher prevalence of ASD among children born pre-term [111,112,113,114,115,116]. Though the risk has been shown to change depending on preterm gestational week cutoff [117]. Though, our study did not find any association with a diagnosis for Early or Threatened labor (ICD-9 642), a common diagnosis made at the discretion of medical personnel.

Other maternal prescriptions resulted in an increased risk of ASD, specifically Anti-inflammatory Glucocorticoids during the first trimester, OR 1.44 (1.08, 1.89), and Respiratory Antihistamines during the second and third trimester, OR 1.39 (1.03, 1.83) and 1.62 (1.20, 2.15), respectively, shown in Table 8. These prescriptions were common treatments for maternal immune dysfunctions and allergens, respectively. However, this study is limited to antihistamine prescriptions that were prescribed and cannot take into account any over-the-counter remedies. Many reviews have shown that maternal inflammatory events in conjunction with maternal immune activation or autoimmune diseases are associated with ASD [8,12,102,118]. Specifically, anti-inflammatory glucocorticoids are a common treatment for psoriasis, which was found to be significantly associated with ASD in one case-control study [17], although a diagnosis for psoriasis (ICD-9 696) was not included in this study due to lack of claims.

Multiple Gestation was found to have a significant increased risk on ASD in all three trimesters (Table 8). However, the study was unable to determine if the twins identified are monozygotic or dizygotic. Previous studies have shown that multiple births have not been associated with ASD, instead the association can be explained by the higher rate of ASD in monozygotic twins compared to their siblings [119,120].

This study does have limitations; the diagnostic codes inputted into each claim were made at the discretion of the medical personnel and were subject to potential bias and all pharmacy claims represented prescription being filled. All variables investigated originate from maternal claims received through insurance and thus does not provide a full representation of all environmental factors that occur outside of insurance claims such as over-the-counter medicines or supplements. Paternal claims were not able to be identified and therefore their influence is unknown. The claims investigated occurred during each woman’s pregnancy and thus do not consider pre-existing conditions that may have been diagnosed or treated prior. Lastly, being limited to claims during pregnancy also ignores the possibility of attenuating these factors through consistent proper treatment during or even after pregnancy.

## 5. Conclusions

Some environmental effects that influence the development of ASD might be identifiable as early as the gestational period. This study identified maternal medical claims made throughout women’s pregnancies and determined risk factors associated with having a child diagnosed with ASD. Identifying these factors that either increase or decrease risk is essential especially for women who are at high risk of having children diagnosed with ASD. It is also beneficially for the child by allowing for early screening, leading to earlier diagnosis and the start of interventions. Early intervention is crucial in children with ASD and has been shown to save costs in the long-term [121,122]. Future research would benefit from exploring medical claims made throughout an individual’s lifetime to truly evaluate health trends and their influence on the risk of having a child with ASD. It would also be of interest to investigate paternal medical claims to emphasize the genetic influence in the development of ASD.

## Figures and Tables

**Figure 1 jpm-11-00950-f001:**
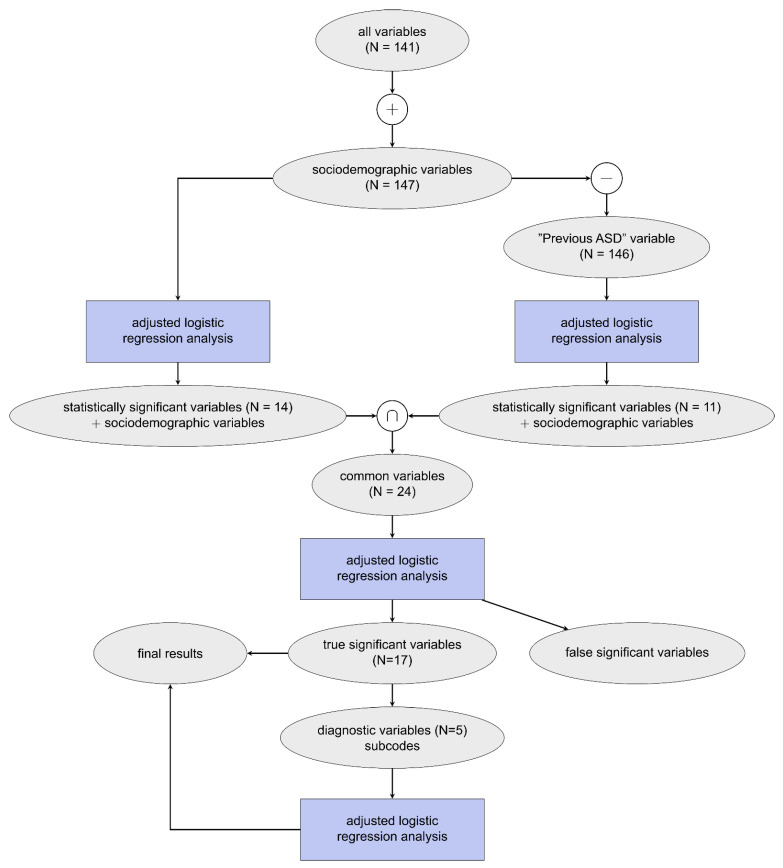
Schematic of model development where N represents the number of variables included for each analysis on claims data obtained throughout the entire pregnancy. This procedure was also repeated for each trimester individually (not shown).

**Figure 2 jpm-11-00950-f002:**
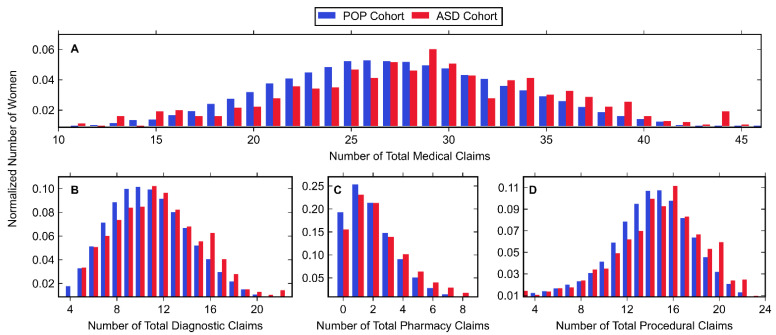
Histogram data of normalized number of women belonging to the POP and ASD cohorts (shown in blue and red, respectively) with (**A**) any medical claim, (**B**) diagnostic claim, (**C**) pharmacy claim, or (**D**) procedural claim. Values associated with small cell sizes are not shown in order to be compliant with OptumLabs’ de-identification policy.

**Figure 3 jpm-11-00950-f003:**
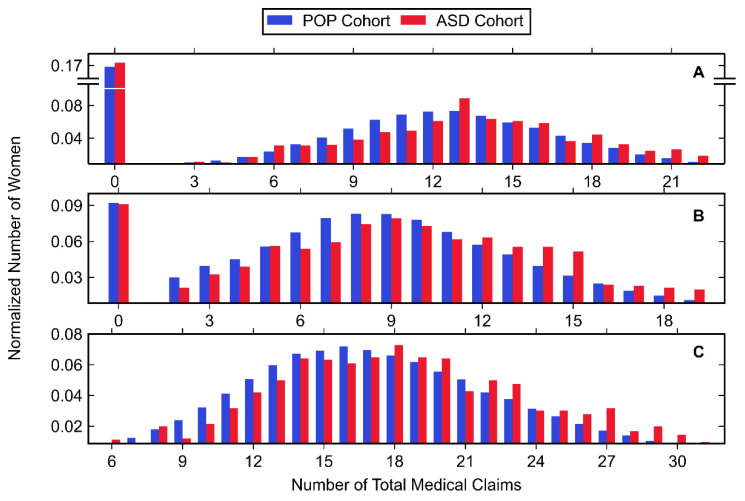
Histogram data of normalized number of women belonging to the POP and ASD cohorts (shown in blue and red, respectively) with any medical claim during the (**A**) first trimester, (**B**) second trimester, and (**C**) third trimester. Values associated with small cell sizes are not shown in order to be compliant with OptumLabs’ de-identification policy.

**Figure 4 jpm-11-00950-f004:**
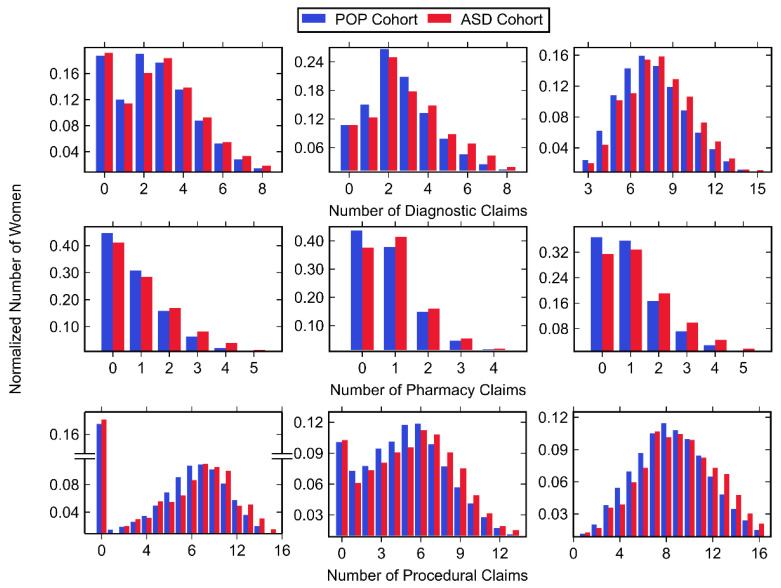
Histogram data of normalized number of women belonging to the POP and ASD cohorts (shown in blue and red, respectively) with any diagnostic (top row), pharmacy (middle row), and procedural (bottom row) claim during the first trimester (left column), second trimester (middle column), and third trimester (right column). Values associated with small cell sizes are not shown in order to be compliant with OptumLabs’ de-identification policy.

**Table 1 jpm-11-00950-t001:** Data Attrition Steps for Identifying Mother Cohorts and Associated Children.

Data Attrition Step	Number of Women	Number of Children
1. Women who have delivery claim between 2000 and 2010	1,241,757	--
2. Women between the ages of 15 and 49, correct medical coverage, and known relationship ID	1,023,631	--
3. Children cohort identified by protocol in Vargason et al. 2019	--	283,644
4. Children with relationship ID as “child” and have a single OLDW family ID	--	234,366
5. Women and children with the same OLDW family ID	133,490	170,480
6. Women who have delivery claim within 10 days of child’s first enrollment date	115,092	136,200
7. Women and children who have single linkage, and women who are continuously enrolled for one year prior to delivery claim	115,069	136,178
8. Women who have sociodemographic information on file	104,051	123,824

**Table 2 jpm-11-00950-t002:** Progression of the Number of Variables for Diagnostic, Pharmacy, and Procedural Categories.

	Number of Variables		
	Diagnostic	Pharmacy	Procedural
Initially investigated with a claim within 10 months of delivery	478	132	122
Contained claims for greater than 2% of women within 10 months of delivery	82	27	45
Contained claims for greater than 2% of women within identified gestational days	77	25	44
Contained claims that were not highly correlated (r ^1^ < 0.7)	76	23	42

^1^ Pearson correlation coefficient.

**Table 3 jpm-11-00950-t003:** Sociodemographic Data on Women Cohorts.

		ASD Women		POP Women		
		1218		102,833		
		*n* ^1^	%	*n* ^1^	%	*p*-Value ^2^
Race	Asian	127	10.4	7949	7.7	**<0.001**
	Black	91	7.5	8275	8.0	0.50
	Hispanic	119	9.8	11,088	10.8	0.28
	White	881	72.3	75,521	73.4	0.40
Home Ownership	Unknown	>118	>9.6	10,304	10.0	0.61
	Probable Owner	1089	89.4	92,117	89.6	0.88
	Probable Renter ^3^	<11	<1.0	412	0.4	0.10
Education Level	Less than 12th Grade ^3^	<11	<1.0	305	0.3	0.79
	High School Diploma	>153	>12.4	19,160	18.6	**<0.001**
	Less than a Bachelor’s Degree	666	54.7	54,339	52.8	0.21
	Bachelor’s Degree Plus	388	31.9	29,029	28.2	**0.006**
Income Range	<USD 40,000	79	6.5	7841	7.6	0.15
	USD 40,000–74,999	206	16.9	19,776	19.2	**0.045**
	USD 75,000–124,999	347	28.5	31,837	31.0	0.07
	USD 125,000–199,999	316	25.9	23,786	23.1	**0.023**
	>USD 200,000	270	22.2	19,593	19.1	**0.007**
Age Range	<20 ^3^	<11	<1.0	290	0.3	0.27
	20–29	312	25.6	32,587	31.7	**<0.001**
	30–39	805	66.1	63,991	62.2	**0.006**
	40–49	>90	>7.3	5965	5.8	**<0.001**
Previous ASD	Yes	245	20.1	0	0.0	**<0.001**
	No	973	79.9	102,833	100.0	**<0.001**

^1^ Number of women. ^2^ *p*-values are calculated using chi-squared analysis or Fisher’s exact test for small cell values, significant *p*-values are shown in bold. ^3^ Values < 11 are censored for anonymity and *p*-values are calculated using Fisher’s exact test.

**Table 4 jpm-11-00950-t004:** Age Statistics on Women Cohorts.

		ASD Women	POP Women	
		Mean (Median)		*p*-Value ^1^
Age Range	<20 ^2^			
	20–29	26.8 (27)	26.6 (27)	0.36
	30–39	34.1 (34)	33.8 (34)	**<0.001**
	40–49	42.3 (42)	42.3 (42)	0.80

^1^ *p*-values are calculated using chi-squared analysis or Fisher’s exact test for small cell values, significant *p*-values are shown in bold. ^2^ values < 11 are censored for anonymity and *p*-values are calculated using Fisher’s exact test.

**Table 5 jpm-11-00950-t005:** Statistics on Number of Medical Claims during Entire Pregnancy.

							Difference in Distribution	
							Variance	Mean
	Cohort	Max	Mean	Median	Mode	Stdev ^2^	*p*-Value ^1^ (95% CI ^3^)	*p*-Value ^1^ (95% CI ^3^)
All Medical Claims	POP	73	27.1	27	26	8.6	**<0.001**	**<0.001** ** ^4^ **
	ASD	59	28.6	29	29	9.4	(1.10, 1.40)	(1.13, 2.74)
Diagnostic Claims	POP	34	11.1	11	10	4.0	**<0.001**	**<0.001** ** ^4^ **
	ASD	29	11.7	11	11	4.2	(1.09, 1.40)	(0.34, 1.06)
Pharmacy Claims	POP	15	2.1	2	1	1.8	**<0.001**	**<0.001** ** ^4^ **
	ASD	15	2.5	2	1	2.1	(1.20, 1.54)	(0.29, 0.65)
Procedural Claims	POP	31	13.9	14	15	4.4	0.102	**<0.001** ** ^5^ **
	ASD	26	14.4	15	16	4.7	(0.98, 1.26)	(0.36, 1.18)

^1^ All *p*-values are calculated using random samples of 1000, significant *p*-values are shown in bold. ^2^ Standard deviation. ^3^ Confidence intervals. ^4^ Calculated using Welch two-sample *t*-test. ^5^ Calculated using two-sample *t*-test.

**Table 6 jpm-11-00950-t006:** Statistics of Number of Medical Claims during Each Trimester.

							Difference in Distribution	
							Variance	Mean
	Cohort	Max	Mean	Median	Mode	Stdev ^2^	*p*-Value ^1^ (95% CI ^3^)	*p*-Value ^1^ (95% CI ^3^)
**Trimester 1**								
All Medical Claims	POP	38	10.6	11	0	6.5	**0.040**	**0.011** ** ^4^ **
	ASD	33	11.2	12	0	7.0	(1.01, 1.29)	(0.18, 1.38)
Diagnostic Claims	POP	15	2.7	3	2	2.1	0.080	0.077 ^5^
	ASD	11	2.8	3	0	2.2	(0.99, 1.26)	(−0.01,0.36)
Pharmacy Claims	POP	10	0.9	1	0	1.1	**<0.001**	**<0.001** ** ^4^ **
	ASD	8	1.1	1	0	1.3	(1.19, 1.52)	(0.10, 0.30)
Procedural Claims	POP	21	7.0	8	0	4.2	0.076	**0.038** ** ^5^ **
	ASD	18	7.2	8	0	4.5	(0.99, 1.27)	(0.02, 0.79)
**Trimester 2**								
All Medical	POP	36	8.7	9	0	5.2	**<0.001**	**<0.001** ** ^4^ **
Claims	ASD	36	9.5	9	0	5.7	(1.13, 1.44)	(0.53, 1.47)
Diagnostic	POP	14	2.7	2	2	1.9	**<0.001**	**<0.001** ** ^4^ **
Claims	ASD	12	3.0	3	2	2.1	(1.20, 1.54)	(0.21, 0.56)
Pharmacy	POP	8	0.8	1	0	0.9	**0.022**	**<0.001** ** ^4^ **
Claims	ASD	8	0.9	1	1	1.0	(1.02, 1.31)	(0.06, 0.23)
Procedural	POP	20	5.1	5	6	3.3	**0.019**	**0.002** ** ^4^ **
Claims	ASD	16	5.5	6	6	3.5	(1.02, 1.31)	(0.17, 0.77)
**Trimester 3**								
All Medical	POP	47	17.4	17	16	5.9	0.105	**<0.001** ** ^5^ **
Claims	ASD	42	18.5	18	18	6.1	(0.98, 1.25)	(0.75, 1.82)
Diagnostic	POP	21	7.7	7	7	2.6	0.420	**0.002** ** ^5^ **
Claims	ASD	17	8.1	8	8	2.6	(0.93, 1.19)	(0.13, 0.59)
Pharmacy	POP	9	1.1	1	0	1.1	**<0.001**	**<0.001** ** ^4^ **
Claims	ASD	8	1.3	1	1	1.3	(1.25, 1.60)	(0.18, 0.40)
Procedural	POP	25	8.6	8	8	3.6	0.537	**<0.001** ** ^5^ **
Claims	ASD	21	9.1	9	7	3.8	(0.92, 1.18)	(0.31, 0.97)

^1^ All *p*-values are calculated using random samples of 1000, significant *p*-values are shown in bold. ^2^ Standard deviation. ^3^ Confidence intervals. ^4^ Calculated using Welch two-sample *t*-test. ^5^ Calculated using two-sample *t*-test.

**Table 7 jpm-11-00950-t007:** Logistic Regression Analysis Results of Variables Identified as Highly Significant during the Entire Pregnancy.

	Odds Ratio ^1^ (95% CI ^2^)	*p*-Value ^1^
Sociodemographic Variables		
Previous ASD	16.1 (8.27, 32.1)	<0.001
Race—Asian	1.36 (1.11, 1.66)	0.003
**Diagnostic Variables**		
Special Screening for Blood Disorders	1.39 (1.06, 1.80)	0.015
Other Indications Related to Labor	1.28 (1.13, 1.45)	<0.001
Other Complications of Labor	1.27 (1.09, 1.47)	0.002
Other Conditions Complicating Pregnancy	0.70 (0.54, 0.88)	0.003
Special Screening for Malignant Neoplasms	0.79 (0.66, 1.00)	0.005
**Pharmacy Variables**		
Antidepressants	1.44 (1.15, 1.79)	0.001
Durable Medical Equipment Diabetic	1.27 (1.00, 1.59)	0.043
Nutritional Vitamins	1.18 (1.04, 1.33)	0.008
**Procedural Variables**		
Anesthesia Procedures Lower Abdomen	1.81 (1.30, 2.46)	<0.001
Procedures Pulmonary	1.66 (1.24, 2.19)	<0.001
Procedures Services Psychiatry	1.44 (1.17, 1.76)	<0.001
Services Consultation	1.37 (1.21, 1.56)	<0.001
Procedures Diagnostic Radiology	1.15 (1.02, 1.30)	0.025
Vaccinations	0.58 (0.46, 0.71)	<0.001
Evaluations Physical Medicine and Rehabilitation	0.77 (0.62, 0.94)	0.015

^1^ *p*-value and odds ratios are calculated using adjusted logistic regression analysis. ^2^ Confidence intervals.

**Table 8 jpm-11-00950-t008:** Logistic Regression Analysis Results of Variables Identified as Highly Significant during each Trimester.

	Trimester 1	Trimester 2	Trimester 3
	Odds Ratio ^1^ (95% CI ^2^) *p*-Value ^1^	Odds Ratio ^1^ (95% CI ^2^) *p*-Value ^1^	Odds Ratio ^1^ (95% CI ^2^) *p*-Value ^1^
**Diagnostic Variables**			
Multiple Gestation	1.49 (1.10, 1.98) 0.009	1.69 (1.33, 2.12) <0.001	1.56 (1.25, 1.94) <0.001
Other Disorders of Urethra and Urinary Tract	1.49 (1.19, 1.86) <0.001	--	--
Other Symptoms Involving Abdomen and Pelvis	1.42 (1.08, 1.83) 0.010	1.42 (1.09, 1.82) 0.007	--
Special Screening for Endocrine Nutritional Metabolic and Immunity Disorders	1.36 (1.05, 1.73) 0.016	--	--
Normal Pregnancy	--	1.33 (1.16, 1.54) <0.001	--
Acquired Hypothyroidism	--	--	1.31 (0.99, 1.70) <0.050
Other Indications Related to Labor	--	1.33 (1.14, 1.54) <0.001	1.27 (1.13, 1.43) <0.001
Other Complications of Labor	--	--	1.22 (1.05, 1.42) 0.010
Need for Prophylactic Vaccination against Certain Viral Diseases	--	0.60 (0.41, 0.83) 0.004	--
Antepartum Hemorrhage and Placenta Previa	--	--	0.71 (0.55, 0.92) 0.010
Other Conditions Complicating Pregnancy	--	--	0.74 (0.55, 0.97) 0.036
Abnormality of Pelvis	0.64 (0.44, 0.92) 0.019	--	0.86 (0.75, 0.99) 0.044
Special Screening for Malignant Neoplasms	0.76 (0.63, 0.91) 0.004	--	--
Antenatal Screening	0.82 (0.71, 0.95) 0.007	--	--
Special Investigations and Examinations	0.83 (0.73, 0.95) 0.006	--	--
**Pharmacy**			
Anti-inflammatory Glucocorticoids	1.44 (1.08, 1.89) 0.011	--	--
Antidepressants	1.40 (1.08, 1.81) 0.010	1.45 (1.08, 1.93) 0.011	1.55 (1.18, 2.00) 0.001
Respiratory Antihistamines	--	1.39 (1.03, 1.83) 0.026	1.62 (1.20, 2.15) 0.001
Nutritional Vitamins	--	1.19 (1.05, 1.34) 0.005	1.19 (1.06, 1.34) 0.004
**Procedural Variables**			
Consultations Clinical Pathology	1.76 (1.25, 2.45) 0.001	--	--
Procedures Services Psychiatry	1.66 (1.24, 2.19) <0.001	1.64 (1.20, 2.20) 0.001	1.46 (1.13, 1.85) 0.003
Surgical Procedures Female Genital System	1.65 (1.26, 2.15) <0.001	--	--
Services Consultation	1.56 (1.30, 1.87) <0.001	--	1.48 (1.25, 1.75) <0.001
Surgical Procedures Nervous System	--	--	1.32 (1.00, 1.72) 0.045
Surgical Procedures Maternity Care and Delivery	--	1.31 (1.09, 1.57) 0.004	--
Services Office or Other Outpatient	--	1.23 (1.08, 1.40) 0.002	--
Procedures Urinalysis	--	--	1.18 (1.04, 1.33) 0.012
Procedures Diagnostic Ultrasound	--	0.61 (0.51, 0.73) <0.001	--
Evaluations Physical Medicine and Rehabilitation	--	0.67 (0.48, 0.92) 0.016	--
Vaccinations	--	--	0.68 (0.47, 0.96) 0.037
Procedures Other Pathology and Laboratory	0.79 (0.65, 0.96) 0.022	--	--

^1^ *p*-value and odds ratios are calculated using adjusted logistic regression analysis. ^2^ Confidence intervals.

**Table 9 jpm-11-00950-t009:** Summary of Study Findings.

Finding	Associated Risk of ASD
Being of Asian race	Increase
Having previous children diagnosed with ASD	Increase
First pregnancy over the age of 35	Increase
Current cesarean delivery	Increase
Prescription for antidepressants and procedure of psychiatric services	Increase
Pre-existing diabetes	Increase
Pre-natal vitamins	Increase
Vaccinations	Decrease
Standard obstetrical procedures	Decrease
Non-pregnancy related procedures	Increase
Urinary tract infection during the first trimester	Increase
Premature pregnancy	Increase
Maternal immune dysfunction or allergens	Increase
Multiple gestation	Increase

## Data Availability

Restrictions apply to the availability of these data. OptumLabs carefully manages access to its data to ensure appropriate use in accordance with its mission and values, policies and procedures, and prevention of re-identification. Users may, therefore, only access OLDW under an agreement with OptumLabs, and compliance with this policy.

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
