# Peer review of "Pregnant Mothers’ Medical Claims and Associated Risk of Their Children being Diagnosed with Autism Spectrum Disorder"

_jpm, 2021, doi:10.3390/jpm11100950_

Round 1
Reviewer 1 Report
The manuscript titled “Pregnant Mothers’ Medical Claims and Associated Risk of their Children being Diagnosed with Autism Spectrum Disorder (ASD)” aimed to identify mothers’ medical events and risks associated with children’s diagnosis of autism spectrum disorder during pregnancy. The results revealed significant relationships between several maternal medical claims (having a biological sibling with ASD, maternal use of an antidepressant, psychiatry services, and claims not related to pregnancy etc.) and child’s diagnosis of ASD. This is an important study that helps us obtain more detailed understanding associated with environmental risk factors of ASD. However, there seen to be a number of points that should be reconsidered in the manuscript.
Major Comment #1.
In the Materials and Methods section, Figure 1 doesn't seem to be so important. Since the number of charts is large, would you consider to put it in the supplementary file?
Major Comment #2.
Mention IRB and add the certification number in the Methods.
Major Comment #3.
Figure 3 looks wrong.
Major Comment #4.
Table 5 can generally be understood by the description in the text. Put it in the supplementary file.
Major Comment #5.
The authors described in the Results section as follows: “For all trimesters, the ASD cohort had a significantly higher mean number of diagnostics, pharmacy and procedural claims with one exception: diagnostic claims made in the first trimester showed no significant difference.” However, the distribution mean of diagnostic claims in trimester 1 (0.077) is in bold in Table 7. Why is the number bolded?
Major Comment #6.
The authors describe in the first paragraph of the Discussion section as follows: “Many of these correspond to a single variable in the model such as having a previous child with ASD, first pregnancy over the age of 35, current cesarean delivery, prescription for antidepressants, psychiatric services, pre-existing diabetes, urinary tract infection during the first trimester, and premature pregnancy.” As the underlined part (first pregnancy…delivery) does not appear in the results, readers may be confused. The readers can understand this meaning by reading the latter part. It is preferable to edit this part.
Major Comment #7.
The authors describe in the Discussion section as follows: “Therefore, it is most likely that the increase in risk associated with Asian race found in this study was a result of sample bias.” Why did the sampling bias occur? Explain it.
Major Comment #8.
The data showed conflicting findings that the current cesarean delivery increased the risk of ASD and a previous cesarean delivery decreased the risk of ASD. The authors discussed it, as follows: "It is possible that women with previous cesarean deliveries may be more aggressive in their preventative and overall prenatal care." Show any evidence for it. This sentence seems to be an overstatement.
Major Comment #9.
The authors showed that diabetes was associated with ASD. However, it is possible that the characteristics of individuals predisposed to diabetes acts as a confounding factor.
Major Comment #10.
The study revealed that standard obstetrical procedures showed a decreased risk. The authors described in the Discussion section as follows: “These findings may suggest that women with earlier and more aggressive obstetrical care have a decreased risk of ASD.” Are there any biological hypotheses of it? Are there any previous studies that support this result?
Major Comment #11.
Why did the urinalysis increase the risk of ASD although the diagnosis of preeclampsia was not significantly different? Explain it (Line 491-498).
Major Comment #12.
Why are the results of premature delivery inconsistent with previous studies? Clarify it (Line 509-510).
Major Comment #13.
Although the study was conducted with many subjects, there are some limitations due to the use of medical claims and some results are inconsistent with previous studies. Clarify the aim of this study again.
Minor Comment #1.
The authors describe in the Materials and Methods section as follows: “Vargason et al. (2019) used the children’s diagnostic claims from their date of birth until five years of age to identify children diagnosed with ASD.” It is better to show the citation number at the end of the sentence.
Minor Comment #2.
In the Materials and Methods section, “Of the original 732 variables investigated, 156 remained after thresholding: 82 ICD-9 diagnostic codes, 29 pharmacy variables, and 45 procedural variables, see Table 2.” The number of Pharmacy variables does not match that in Table 2 (27). Correct it.
Major Comment #3.
In the Discussion section, “The data cohorts identified in this study found the highest percentage of ASD pregnancies among White children, followed by Asian, Black and Hispanic.” It may be a mistake for "White children, followed by Asian, Hispanic and Black."
Author Response
REVIEWER 1 – COMMENTS TO THE AUTHOR
Summary
The manuscript titled “Pregnant Mothers’ Medical Claims and Associated Risk of their Children being Diagnosed with Autism Spectrum Disorder (ASD)” aimed to identify mothers’ medical events and risks associated with children’s diagnosis of autism spectrum disorder during pregnancy. The results revealed significant relationships between several maternal medical claims (having a biological sibling with ASD, maternal use of an antidepressant, psychiatry services, and claims not related to pregnancy etc.) and child’s diagnosis of ASD. This is an important study that helps us obtain more detailed understanding associated with environmental risk factors of ASD. However, there seen to be a number of points that should be reconsidered in the manuscript.
Author Response: We thank the reviewer for their detailed summary of our manuscript and have addressed all his/her comments below.
Major Comment #1: In the Materials and Methods section, Figure 1 doesn't seem to be so important. Since the number of charts is large, would you consider to put it in the supplementary file?
Author Response #1: We agree with the reviewer and have moved Figure 1 to the supplementary information.
Major Comment #2: Mention IRB and add the certification number in the Methods.
Author Response #2: Since the study uses deidentified data, it is exempt from IRB review per Code of Federal Regulations statute 45 CFR 46.104(d)(4). Thus, IRB approval was not needed for this study; this has now been stated in the Methods section.
Major Comment #3: Figure 3 looks wrong.
Author Response #3: We greatly appreciate the reviewer bringing this to our attention and have corrected Figure 3.
Major Comment #4: Table 5 can generally be understood by the description in the text. Put it in the supplementary file.
Author Response #4: We thank the reviewer and agree with this comment; we have moved Table 5 to the supplemental information.
Major Comment #5: The authors described in the Results section as follows: “For all trimesters, the ASD cohort had a significantly higher mean number of diagnostics, pharmacy and procedural claims with one exception: diagnostic claims made in the first trimester showed no significant difference.” However, the distribution mean of diagnostic claims in trimester 1 (0.077) is in bold in Table 7. Why is the number bolded?
Author Response #5: We thank the reviewer for bringing this typo to our attention and have ensured all non-significant p-values are not bolded in all tables.
Major Comment #6: The authors describe in the first paragraph of the Discussion section as follows: “Many of these correspond to a single variable in the model such as having a previous child with ASD, first pregnancy over the age of 35, current cesarean delivery, prescription for antidepressants, psychiatric services, pre-existing diabetes, urinary tract infection during the first trimester, and premature 3 pregnancy.” As the underlined part (first pregnancy…delivery) does not appear in the results, readers may be confused. The readers can understand this meaning by reading the latter part. It is preferable to edit this part.
Author Response #6: We thank the reviewer for this comment and have added the referred variables in parenthesis to reduce confusion in this sentence.
Major Comment #7: The authors describe in the Discussion section as follows: “Therefore, it is most likely that the increase in risk associated with Asian race found in this study was a result of sample bias.” Why did the sampling bias occur? Explain it.
Author Response #7: As mentioned in the results section, our study found a statistically significant greater number of Asian women belonging to the ASD cohort compared to the POP cohort. We thank the reviewer for bringing this to our attention and have addressed this statement in the discussion section.
Major Comment #8: The data showed conflicting findings that the current cesarean delivery increased the risk of ASD and a previous cesarean delivery decreased the risk of ASD. The authors discussed it, as follows: "It is possible that women with previous cesarean deliveries may be more aggressive in their preventative and overall prenatal care." Show any evidence for it. This sentence seems to be an overstatement.
Author Response #8: Unfortunately, there is little research on the effect of previous cesarean delivery or even vagina birth after cesarean (VBAC) and the effect on having a child diagnosed with ASD. Women with previous cesarean deliveries are more carefully managed, especially during labor (Knight et al., 2014; Lydon-Rochelle et al., 2010). This had been addressed in the discussion section.
Major Comment #9: The authors showed that diabetes was associated with ASD. However, it is possible that the characteristics of individuals predisposed to diabetes acts as a confounding factor.
Author Response #9: We thank the reviewer for this comment and have addressed this potential confounding factor in the discussion section.
Major Comment #10: The study revealed that standard obstetrical procedures showed a decreased risk. The authors described in the Discussion section as follows: “These findings may suggest that women with earlier and more aggressive obstetrical care have a decreased risk of ASD.” Are there any biological hypotheses of it? Are there any previous studies that support this result?
Author Response #10: It has been well documented that obstetric complications increase the risk of having a child with ASD (Burstyn et al., 2010; Dodds et al., 2011; Lyall et al., 2012) and it is clinically recommended that women who are at risk of having a child with ASD should be closely monitored throughout their pregnancy (Emberti Gialloreti et al., 2019). Our findings support these claims, which we have clarified in the discussion section.
Major Comment #11: Why did the urinalysis increase the risk of ASD although the diagnosis of preeclampsia was not significantly different? Explain it (Line 491-498).
Author Response #11: Urinalysis procedures refer to examination of urine for diagnosing various diseases such as urinary tract infection, pre-eclampsia, and others. Unfortunately, this study cannot determine what the purpose of this procedure is for, only that the procedure had been done. This point has now been clarified in this section.
Major Comment #12: Why are the results of premature delivery inconsistent with previous studies? Clarify it (Line 509-510).
Author Response #12: Pre-term delivery results were consistent with previous studies, as stated in this section. Our study did not find any significance associated with the diagnosis of Early or Threatened Labor (ICD-9 642) which is a common diagnosis for pre-term children. This may be due to the discrepancies of claims inputted by medical personnel (a limitation listed in our conclusion).
Major Comment #13. Although the study was conducted with many subjects, there are some limitations due to the use of medical claims and some results are inconsistent with previous studies. Clarify the aim of this study again.
Author Response #13: We thank the reviewer for this comment and have reiterated the aim of the study in the conclusion section.
Minor Comment #14: The authors describe in the Materials and Methods section as follows: “Vargason et al. (2019) used the children’s diagnostic claims from their date of birth until five years of age to identify children diagnosed with ASD.” It is better to show the citation number at the end of the sentence.
Author Response #14: We thank the reviewer for this comment. The citation number is at the end of the sentence prior to the one mentioned by the reviewer, where the study is first mentioned.
Minor Comment #15: In the Materials and Methods section, “Of the original 732 variables investigated, 156 remained after thresholding: 82 ICD-9 diagnostic codes, 29 pharmacy variables, and 45 procedural variables, see Table 2.” The number of Pharmacy variables does not match that in Table 2 (27). Correct it.
Author Response #15: We thank the reviewer to identifying this typo and have corrected it.
Minor Comment #16: In the Discussion section, “The data cohorts identified in this study found the highest percentage of ASD pregnancies among White children, followed by Asian, Black and Hispanic.” It may be a mistake for "White children, followed by Asian, Hispanic and Black."
Author Response #16: We thank the reviewer for bringing this to our attention and have corrected this sentence.
Citations:
Burstyn, I., Sithole, F., & Zwaigenbaum, L. (2010). Autism spectrum disorders, maternal characteristics and obstetric complications among singletons born in Alberta, Canada. Chronic Diseases in Canada, 30(4), 125–134.
Dodds, L., Fell, D. B., Shea, S., Armson, B. A., Allen, A. C., & Bryson, S. (2011). The Role of Prenatal, Obstetric and Neonatal Factors in the Development of Autism. Journal of Autism and Developmental Disorders, 41(7), 891–902. https://doi.org/10.1007/s10803-010-1114-8
Emberti Gialloreti, L., Mazzone, L., Benvenuto, A., Fasano, A., Alcon, A. G., Kraneveld, A., Moavero, R., Raz, R., Riccio, M. P., Siracusano, M., Zachor, D. A., Marini, M., & Curatolo, P. (2019). Risk and Protective Environmental Factors Associated with Autism Spectrum Disorder: Evidence-Based Principles and Recommendations. Journal of Clinical Medicine, 8(2), 217. https://doi.org/10.3390/jcm8020217
Knight, H., Gurol-Urganci, I., van der Meulen, J., Mahmood, T., Richmond, D., Dougall, A., & Cromwell, D. (2014). Vaginal birth after caesarean section: A cohort study investigating factors associated with its uptake and success. BJOG: An International Journal of Obstetrics & Gynaecology, 121(2), 183–192. https://doi.org/10.1111/1471-0528.12508
Lyall, K., Pauls, D. L., Spiegelman, D., Ascherio, A., & Santangelo, S. L. (2012). Pregnancy complications and obstetric suboptimality in association with autism spectrum disorders in children of the nurses’ health study II. Autism Research, 5(1), 21–30. https://doi.org/10.1002/aur.228
Lydon-Rochelle, M. T., Cahill, A. G., & Spong, C. Y. (2010). Birth After Previous Cesarean Delivery: Short-Term Maternal Outcomes. Seminars in Perinatology, 34(4), 249–257. https://doi.org/10.1053/j.semperi.2010.03.004
Reviewer 2 Report
Thank you very much for the opportunity to read and review the paper entitled: „Pregnant Mothers’ Medical Claims and Associated Risk of their Children being Diagnosed with Autism Spectrum Disorder“.
This is a retrospective study examining risk factors associated with Autism Spectrum Disorder. This is an impressive study and I commend the authors for the great work.
I only have a couple of minor remarks regarding this paper.
- In the first paragraph of the Introduction, the authors should state the ratio of prevalence in relation to gender and provide a reference.
- Page 2, line 60, can you please tell us which cognitive and mental disorders so we do not have to search the references for it
- Did the Ethical Review Board of authors' institutions approve the study? Please make this clear.
- I do not think the claim that food supplements in the form of nutritional vitamins increase the risk of ASD is valid. The reason for this is that the authors did not examine which particular vitamins were used by mothers. You cannot simply put all the vitamins in a single category and examine the risk. If the authors do not have data on which vitamins were used, then the claim is not a valid one. All other factors seem to be more precisely defined.
Round 2
Reviewer 1 Report
The authors has properly revised the manuscript according to the comments. This paper is suitable for publication.